# Synthesis of a Polyaniline Nanoparticle Using a Solution Plasma Process with an Ar Gas Bubble Channel

**DOI:** 10.3390/polym11010105

**Published:** 2019-01-09

**Authors:** Jun-Goo Shin, Choon-Sang Park, Eun Young Jung, Bhum Jae Shin, Heung-Sik Tae

**Affiliations:** 1School of Electronics Engineering, College of IT Engineering, Kyungpook National University, Daegu 41566, Korea; bmw345@ee.knu.ac.kr (J.-G.S.); purplepcs@ee.knu.ac.kr (C.-S.P.); eyjung@knu.ac.kr (E.Y.J.); 2Department of Electronics Engineering, Sejong University, Seoul 05006, Korea; hahusbj@sejong.ac.kr

**Keywords:** solution plasma, polyaniline nanoparticle, gas bubble channel, polymerization

## Abstract

This work researched polymerization of liquid aniline monomer by solution plasma with a gas bubble channel and investigated characteristics of solution plasma and polyaniline (PANI). The injected gas bubble channel in the proposed solution plasma process (SPP) played a significant role in producing a stable discharge in liquid aniline monomer at a low voltage and furthermore enhancing the contact surface area between liquid aniline monomer and plasma, thereby achieving polymerization on the boundary of the liquid aniline monomer and plasma. Solution plasma properties were analyzed with voltage–current, optical emission spectroscopy, and high-speed camera. Conductivity, percentage yield, and firing voltage of PANI nanoparticle dispersed solution were measured. To investigate the characteristics of synthesized PANI nanoparticles, field emission scanning electron microscopy, dynamic light scattering, transmission electron microscopy, selective area electron diffraction (SAED) pattern, Fourier transform infrared spectroscopy (FTIR), gel permeation chromatography, ^1^H-nuclear magnetic resonance (^1^H-NMR), and X-ray photo spectroscopy (XPS) were examined. The FTIR, ^1^H-NMR, and XPS analysis showed the PANI characteristic peaks with evidence that some quinoid and benzene rings were broken by the solution plasma process with a gas bubble channel. The results indicate that PANI nanoparticles have a spherical shape with a size between 25 and 35 nm. The SAED pattern shows the amorphous pattern.

## 1. Introduction

Many scientists or engineers have investigated many considerable advances and changes in the synthesis and characterization of conducting polyaniline (PANI) for the last sixty years [1,2,3,4,5,6,7]. PANI nanoparticles have received great attention due to their potential applications in optoelectronics, chemical sensors, electromagnetic shielding, molecular electronics, and bionanotechnologies. Conventionally, PANI nanoparticles have been synthesized via chemical, electrochemical, electrospinning, seeding polymerization, ultrasonic assisted polymerization, hard and soft templates, and interfacial polymerization methods [8,9,10,11,12,13,14,15,16,17,18,19,20]. Among these methods, the electrochemical solution plasma process (SPP) is a new, simple, and useful synthesis method of polymer nanoparticles, carbon nanomaterials, metal nanoparticles, surface treatment of polymers, purification of wastewater, and degradation of organic compounds, because this nonequilibrium plasma can provide rapid reaction and synthesis due to the reactive chemical species and radicals [21,22,23,24,25,26,27]. Electrical discharge in the solution phase is known as “SPP” and has recently been studied to be an effective synthesis method. In addition, SPP does not include any strong chemical reagents. Therefore, the removal of chemical residue is not required. However, most plasma materials grown using SPP without a gas bubble channel tend to show poor nanoparticle qualities with weak chemical stabilities, which would inherently result from the use of plasma with unstable discharge in solution conditions. Moreover, it is difficult to obtain polymer nanoparticles with uniform size due to the small contact surface area and unstable discharge. To obtain a uniform size of polymer nanoparticles via the conventional SPP method, it is very important to increase the contact surface area and to produce the stable discharge [28,29]. Therefore, in this work, we propose an advanced SPP with an injected gas bubble channel to increase the contact surface area and to produce a stable discharge during the electrochemical plasma synthesis of PANI nanoparticles from the aniline monomer solution in SPP devices. A gas bubble channel is often used to enhance the solution plasma performance in pulsed discharge systems. The injection of gas bubbles can significantly enhance the contact surface area of the gas and monomer solution and increase higher-energy electron production. In addition, gas bubbles can increase the mass transfer rate and can enhance the efficiency of the diffusion of reactive species into the solution [30]. Consequently, the advanced SPP with an Ar gas bubble channel can produce stable plasma in the fragmentation region, and as such increase the mass transfer rate and efficiency of the diffusion of reactive species between the gas and monomer solution. The gas channel formation and related plasma discharge are monitored by using a high speed camera, voltage–current probes, and optical emission spectroscopy (OES). The firing voltage, percentage yield, and conductivity of the monomer solution with PANI nanoparticles synthesized via the advanced SPP are also measured. To characterize the PANI nanoparticles synthesized by the advanced SPP with an Ar gas bubble channel, field emission-scanning electron microscopy, dynamic light scattering, transmission electron microscopy, Fourier transform infrared, ^1^H nuclear magnetic resonance, gel permeation chromatography, and X-ray photoelectron spectroscopy are also used.

## 2. Materials and Methods

### 2.1. Experimental Setup

The plasma reactor consisted of a glass cylinder with an outer diameter (O.D.) and inner diameter (I.D.) of 40 and 34 mm, respectively, and a height of 150 mm. In the two tungsten electrodes, the diameter of each electrode was 0.5 mm, and the distance between the tungsten electrodes was 1 mm. The I.D. of the capillary glass tube for the Ar gas channel was 3 mm. The argon (Ar) gas was injected along with two tungsten electrodes in parallel, thus forming a gas bubble channel between two electrodes. The gap of the Ar gas channel between the tungsten electrode and the capillary glass tube was 2.5 mm. The amount of liquid aniline monomer was 26 mL, and the Ar flow rate was 100 standard cubic centimeters per minute (sccm). A bipolar pulse with an amplitude of 16.4 kV and a frequency of 5 kHz was generated by a high voltage amplifier (20/20C-HS, Trek, Inc., Lockport, NY, USA) and pulse generator (AFG-3102, Tektronix Inc., Beaverton, OR, USA). The bipolar pulse duty ratio was 60 μs and the process time for polymerization was 50 min.

### 2.2. Preparation of Polyaniline

The polyaniline (PANI) nanoparticles were synthesized from liquid aniline monomer using a solution plasma process (SPP). After the synthesis process was carried out for 50 min, the PANI nanoparticles were dispersed within processed liquid aniline monomer. For filtration and cleaning of the PANI grown from aniline monomer, processed liquid aniline monomer (5 mL) and ethanol (15 mL) were mixed together. The mixed solution was rotated using a centrifugal separator for 20 min at 10,000 rpm and the by-product, except precipitated PANI, was removed. The distilled water was added to rinse the precipitated PANI and rotated again at the same rotation conditions. Each step mentioned above was repeated twice. The final step was to dry the precipitated PANI nanoparticles at 60 °C for 12 h in an oven. Hereinafter, the synthesized PANI nanoparticles prepared for measurement are called samples or powder samples.

### 2.3. Voltage–Current (V–I) Measurement

To examine discharge characteristics, a high voltage probe (P6015A, Tektronix Inc., Beaverton, OR, USA) was connected to the powered electrode and a current probe (Pearson Elec. Inc., Palo Alto, CA, USA) was connected to the ground electrode.

### 2.4. Optical Emission Spectroscopy

The optical emissions ranging from 300 to 900 nm emitted from the Ar plasma and the decomposed aniline during solution plasma polymerization process were observed using an optical emission spectrometer (OES, Ocean Optics Inc., Dunedin, FL, USA) with a wide area lens.

### 2.5. High Speed Camera

The formation of a gas bubble channel in liquid aniline monomer and related discharge phenomena were investigated with a high-speed camera (Phantom Miro C110, AMETEK, Wayne, NJ, USA). In this case, two lenses (Nikon AF Nikkor 105 mm, 1:2.8 D, Nikon, Tokyo, Japan and Nikon-AF 36 mm DG) were used at 5000 frames per second (fps) with a shutter time of 200 μs. Resolution was 272 × 256.

### 2.6. Scanning Electron Microscopy

The morphology and size of the synthesized PANI nanoparticles were monitored by field emission-scanning electron microscopy (FE-SEM: SU8220, Hitachi Korea Co. Ltd., Seoul, Korea). For measurement, samples were fixed with paste. Energy dispersive X-ray spectroscopy (EDS) (SU8220, Hitachi Korea Co. Ltd., Seoul, Korea) was employed to identify the element composition of samples.

### 2.7. Dynamic Light Scattering

The size distribution and average particle size of PANI nanoparticles ranging from 0.6 nm to 10 μm were determined by dynamic light scattering (DLS, Otsuka Electronics Co. Ltd., Osaka, Japan). Before the measurement of size distribution, the dispersions of PANI nanoparticles were prepared by 20 min sonication in ethanol.

### 2.8. Transmission Electron Microscopy

PANI nanoparticles were also measured using transmission electron microscopy (TEM) for the precise shape of the image and selected area electron diffraction (SAED) for diffraction patterns. This information was taken using a Titan G2 ChemiSTEM CS Probe (FEI Company, Hillsboro, OR, USA) operating at 200 kV. Samples were dispersed in ethanol and obtained with a carbon-coated copper grid. EDS (FEI Company, Hillsboro, OR, USA) was used to identify the components of the sample.

### 2.9. Fourier Transform Infrared Spectroscopy

Fourier transform infrared spectroscopy (FTIR) spectra of the powder sample was conducted with a PerkinElmer spectrometer (Frontier, Waltham, MA, USA). The measurement method was attenuated total reflection-FTIR (ATR-FTIR) and the spectra range was 400–4000 cm^−1^.

### 2.10. Nuclear Magnetic Resonance Spectroscopy

The detailed electronic structures of PANI nanoparticles were obtained by nuclear magnetic resonance spectroscopy (NMR) (Bricker Company, Billerica, MA, USA) that gave a signal to excite the local magnetic field to nuclei of atoms and detected the resultant resonant signal. The detected signal was changed depending on the environment around the atoms. For measurement, the sample was dissolved in deuterated dimethylformamide (DMF-d_7_) solvent.

### 2.11. Gel Permeation Chromatography

Gel permeation chromatography (GPC) was conducted using an Agilent 1200 series instrument (Santa Clara, CA, USA). The sample was dissolved in dimethylformamide (DMF, HPLC, Sigma-Aldrich Co., St. Louis, MO, USA). Polydispersity index (PDI) was performed with polystyrene standard.

### 2.12. X-ray Photoelectron Spectroscopy

XPS was performed to measure the surface components of PANI nanoparticles using a ESCALAB 250Xi instrument (Thermo Scientific Instruments Co., Ltd., Waltham, MA, USA) with a high resolution of 3 μm. The source of XPS was a micro-focusing monochromator of aluminum with a hemispherical energy analyzer with a mean radius of 150 mm and energy range from 0 to 5000 eV

## 3. Results and Discussion

### 3.1. Discharge Properties with Ar Gas Bubble Channel in Liquid Aniline Monomer

Figure 1 shows a schematic diagram of the solution plasma cylinder and measurement equipment used in this study. Discharge without a gas bubble channel was very difficult to produce in liquid aniline monomer. In addition, most particles and materials synthesized using streamer and unstable discharge without a gas bubble channel tended to show poor nanoparticle quality and properties due to the production of wear out or erosion of electrodes. To overcome these problems, Ar gas was injected to the discharge region in liquid aniline monomer between two tungsten electrodes through two capillary glass tubes. The distance between two tungsten electrodes was 1 mm. In the case that the distance between the two tungsten electrodes is more than 1 mm, the initial discharge voltage is very high and the discharge is difficult to form in the discharge region. The solution plasma cylinder was connected with two capillary glass tubes for supplying Ar gas between two tungsten electrodes. Ar gas bubbles flowing through a capillary glass tube formed a gas bubble channel between the two electrodes, thus enabling the discharge to be produced more easily at a low voltage, even in liquid aniline. The distance between the two capillary glass tubes for injection of Ar gas was 3 mm. In the case that the distance between two capillary glass tubes is more than 3 mm, the Ar gas bubble channel from two capillary glass tubes is difficult to form in the discharge region. Applied peak voltage, bipolar rectangular pulse width, and frequency were 16.4 kV, 60 μs, and 5 kHz, respectively. A bipolar pulse width shorter than 60 μs can limit generation of discharge. Applied and corresponding discharge peak voltage and currents were monitored. Optical emissions during discharge were detected with OES, whereas a high speed camera was used to monitor the behavior of gas bubbles and related discharge.

Figure 2 shows the images of gas bubble evolution in liquid aniline monomer with and without discharge, using a high speed camera. For the non-discharge case, these images clearly show that the Ar gas flowing through each capillary glass tube formed a bubble within the liquid aniline, and was combined into a larger bubble, finally forming a gas bubble channel between two tungsten electrodes (Appendix A). This gas bubble channel enveloped two tungsten electrodes and provided a discharge path between the two tungsten electrodes. The blue color in Figure 2b shows the plasma region generated due to the Ar gas bubble channel (Appendix A). Since the rectangular bipolar pulse had a pulse width of 60 μs at a frequency of 5 kHz, the high voltage bipolar pulse was continually applied irrespective of the formation of a gas bubble channel. At t = 6.8 ms, no discharge was observed in liquid aniline monomer because a gas bubble channel was not yet formed despite the fact that the rectangular bipolar pulse was applied. At t = 10.2 ms, however, plasma was observed to be generated in liquid aniline monomer thanks to the formation of a gas bubble channel, as shown in Figure 2b. The gas bubble channel facilitated a production of plasma due to the lowering of a firing voltage in the liquid aniline environment. The resultant plasma discharge was propagated in a direction of the Ar gas bubble channel, implying that the Ar gas bubble channel played a role in providing a discharge path at a relatively low voltage condition.

Figure 3 shows the applied peak voltage, discharge peak voltage, and current measured during the solution plasma process of liquid aniline monomer. In Figure 3a, the applied peak-to-peak voltage between two tungsten electrodes was 16.4 kV under no discharge condition, where the pulse width was 60 μs at a frequency of 5 kHz. Figure 3b shows the discharge peak voltages measured relative to process time from 10 to 50 min at an interval of 10 min when plasma was produced through the gas bubble channel between two tungsten electrodes. As the processing time increased, the corresponding discharge peak voltage also increased from 2 to 5 kV, indicating that it was difficult to produce a discharge with an increase in the process time. In other words, with an increase in the process time, forming a gas bubble channel between two tungsten electrodes was suppressed, and as such, higher firing voltage was required to generate discharge, thereby resulting in an increase in the discharge peak voltage. In Figure 3c, the corresponding discharge currents show the tendency opposite to the discharge voltage, meaning that the decrease in the discharge current would be caused by an increase of resistance in plasma region depending on process time. Figure 4 shows the variations of firing voltage, percent yield, and conductivity of the synthesized PANI nanoparticle dispersed solution in liquid aniline monomer during process times up to 50 min. In Figure 4a, the firing voltage tended to increase with an increase in the process time during process, implying that a gas bubble channel induced by the flow of the Ar gas bubbles was difficult to form, as mentioned in the previous discussion on the tendency of discharge peak voltage with process time. In Figure 4b, the percentage yield of PANI relative to the process time tended to increase, thus inducing the increase in the viscosity, which can affect formation of a gas bubble channel [31]. In Figure 4c, the conductivity of the synthesized PANI nanoparticle dispersed solution in liquid aniline monomer during process times up to 50 min tended to increase. From the results of Figure 4a–c, when the medium spaced between two tungsten electrodes was converted from liquid aniline into PANI nanoparticle dispersed solution due to SPP, both conductivity and viscosity were increased, thereby resulting in increases in the firing voltage. This phenomenon confirms that the plasma discharge in liquid aniline monomer strongly depends on the formation of a gas bubble channel between two tungsten electrodes.

Figure 5 shows the optical emissions measured from plasma discharge in liquid aniline when PANI nanoparticles were synthesized by the solution plasma process with a gas bubble channel. Molecular emission peaks of CN, CH, and C_2_ were produced by the electron impact dissociation of liquid aniline monomer. Swan bands of C_2_ correspond to 471.52, 516.5, and 561 nm. Swan bands of C_2_ are commonly observed in organic material and originate from conjugated carbon of benzene rings of liquid aniline monomer. The CN peaks of 388 nm are intense peaks, which are related to benzene rings of liquid aniline monomer. The CN violet system has two smaller peaks of 360 and 419 nm, indicating an amine group attached to a benzene ring. Hydrogen Balmer line of H_α_ corresponding to 656.3 nm also indicates a dissociation of liquid aniline monomer. Multiple excited Ar lines are observed from 698 to 854 nm due to the Ar gas bubble injected to ignite plasma [32,33,34].

### 3.2. Properties of Synthesized Polyaniline

Figure 6 shows the synthesized and dispersed PANI nanoparticles in solution and dried PANI nanoparticles from the solution. After synthesizing PANI nanoparticles from liquid aniline monomer by solution plasma with a gas bubble channel for 50 min, the color of solution was changed into black, because polyaniline nanoparticles were dispersed in the solution processed by solution plasma with the gas bubble channel, and nanoparticles were not precipitated on the bottom of the bottle. When PANI nanoparticles were filtered and dried, the PANI nanoparticles agglomerated together to form bigger particles.

SEM was used to measure morphology and size of PANI nanoparticles synthesized by solution plasma with gas bubble channel [35,36]. The SEM image in Figure 7a shows that the PANI nanoparticles synthesized have spherical shapes. The size distribution results in Figure 7b show that the nano sizes of PANI nanoparticles are ranged from 25 to 35 nm with a narrow size distribution. The insets in Figure 7a are the results of EDS, which confirm that the PANI nanoparticles involve components of carbon and nitrogen. In particular, the presence of nitrogen in the PANI nanoparticles is confirmed. From EDS results, when adopting the gas bubble channel, the chemical element compositions (wt%) of tungsten on PANI nanoparticles were remarkably reduced from 6.36 to 1.37% compared to no gas bubble channel. These results confirm that the solution plasma process with a gas bubble channel can produce uniform PANI nanoparticles with very small size and can reduce erosion of the electrodes.

The TEM image of Figure 8a shows a single PANI nanoparticle synthesized using solution plasma process with gas bubble channel. Figure 8a is a magnified image of a single PANI nanoparticle for investigating the PANI nanoparticle in detail. A certain crystal structure in the PANI nanoparticle is not observed in the TEM image. The SAED measurement in the inset of Figure 8a confirms that the PANI nanoparticle has an amorphous pattern [37,38]. Figure 8b shows an image of agglomerated PANI nanoparticles. The EDS analysis in the insets of Figure 8b shows that the PANI nanoparticles synthesized have carbon and nitrogen elements, as shown in insets of Figure 7, because PANI nanoparticles are composed of a benzene ring, quinoid ring, and amine group.

Figure 9 shows the FTIR spectra of PANI nanoparticles synthesized by the solution plasma process with a gas bubble channel. FTIR spectra shown in Figure 9 reveal peaks of C=C bending at 684 cm^−1^, C–H bending at 744 cm^−1^, C–H bending at 818 cm^−1^, C–H bending at 872 cm^−1^, C–N stretching aromatic amine at 1261 cm^−1^, C–C stretching aromatic ring at 1493 cm^−1^, C=C stretching conjugated at 1588 cm^−1^, and N–H stretching aliphatic primary amine at 3332 cm^−1^. In particular, N–H stretching aliphatic primary amine at 3332 cm^−1^ is broad due to the amorphous nature of the PANI nanoparticle. [39,40]. Figure 10 shows the ^1^H-NMR spectrum of PANI nanoparticles dissolved in DMF-d_7_ solvent. ^1^H-NMR is a useful method to confirm molecules in detail. Two high intensity peaks between 2–4 ppm are ^1^H-NMR peaks in DMF solvent. ^1^H-NMR peaks in PANI are mainly centered at 7 ppm and 7, 8 ppm due to protons on pheylene and a disubstituted pheylene unit, as shown in the inset of Figure 10. Peaks of primary amine (–NH_2_) connected to a benzene ring neighboring imine (–C=N) nitrogen connected to a quinoid ring is observed at 6.4–6.5 ppm. Three sharp peaks at 7.08, 7.18, and 7.28 reveal NH^+^ that is 1H coupled to 14N confirming the protonated state. The α indicates broad benzene ring photon peaks of PANI nanoparticles at 7.0 ppm and β indicates quinoid ring photon peaks in the range of 7.2–7.9 ppm [41,42]. Table 1 shows the result of GPC. Although PANI nanoparticles synthesized by solution plasma have low molecular weight (Mw) and low molecular number (Mn), the PANI nanoparticles have excellent polydispersity (PDI) characteristics of about 1.19. As a result, PANI nanoparticles using solution plasma with a gas bubble channel were successfully synthesized. This is one of methods suitable for obtaining polymer nanoparticles under liquid monomer [43,44].

The XPS analysis of synthesized PANI nanoparticles using solution plasma with a gas bubble channel was conducted. As shown in Figure 11a, the elements C 1s (at 284.5 eV), N 1s (at 398.4 eV), and O 1s (at 531.8 eV) were measured on the PANI nanoparticles. The XPS spectra of C 1s, N 1s, and O 1s indicates that PANI nanoparticles consist of carbon, nitrogen and oxygen. The atomic percentage of PANI nanoparticle is 83.4% for C 1s, 6.0% for N 1s, and 10.6% for O 1s. Carbon and nitrogen are components of PANI, whereas oxygen is added from atmospheric air after polymerization. In Figure 11b, six carbon-containing component peaks of C 1s at 284.1, 284.6, 285.7, 286.4, 286.9, and 287.5 eV are confirmed to be corresponding to C–C/C–H, C=C, C–N, C–O, C=O, and O=C–O, respectively. Figure 11c shows the N 1s spectra of the PANI nanoparticles where the three peaks centered at 398.7, 399.7, and 401.1 eV correspond to the quinoid imine (–N–), benzenoid amine-like (–NH–) structure, and positively charged nitrogen (NH_2_^+^) [45,46], respectively. Peaks of C 1s, N 1s, and O 1s are summarized in Table 2 and Table 3.

## 4. Conclusions

PANI nanoparticles were successfully synthesized from liquid aniline monomer by solution plasma with a gas bubble channel. Polymerization was achieved at the boundary of the gas bubble channel and liquid monomer aniline. Plasma is very difficult to generate in liquid aniline. To solve this problem, a gas bubble was provided externally during the solution plasma process, such that the plasma discharge was effectively produced to synthesize the PANI nanoparticle from liquid aniline monomer successfully. The FTIR, NMR, and XPS reveal that the synthesized nanoparticles are PANI. The synthesized PANI nanoparticles are observed by SAED to be an amorphous pattern. The SEM, DLS, and TEM confirm that the size of the PANI nanoparticle is estimated to be about tens of nanometers and has a spherical shape.

## Figures and Tables

**Figure 1 polymers-11-00105-f001:**
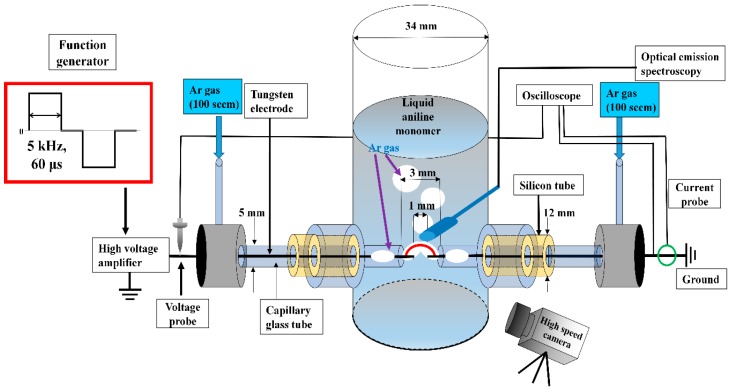
Schematic diagram of the solution plasma process in liquid aniline monomer with the proposed argon (Ar) gas bubble channel and measurement setup used in this study.

**Figure 2 polymers-11-00105-f002:**
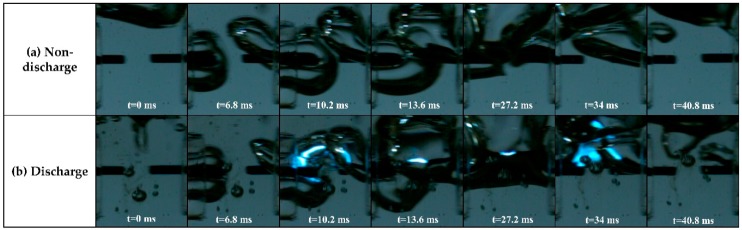
Changes in the gas bubble in (**a**) a non-discharge case without applying a rectangular bipolar pulse, and in (**b**) a discharge case with applying a rectangular bipolar pulse relative to solution plasma process time. These images were obtained by high speed camera with 5000 fps and exposure time of 197 μs.

**Figure 3 polymers-11-00105-f003:**
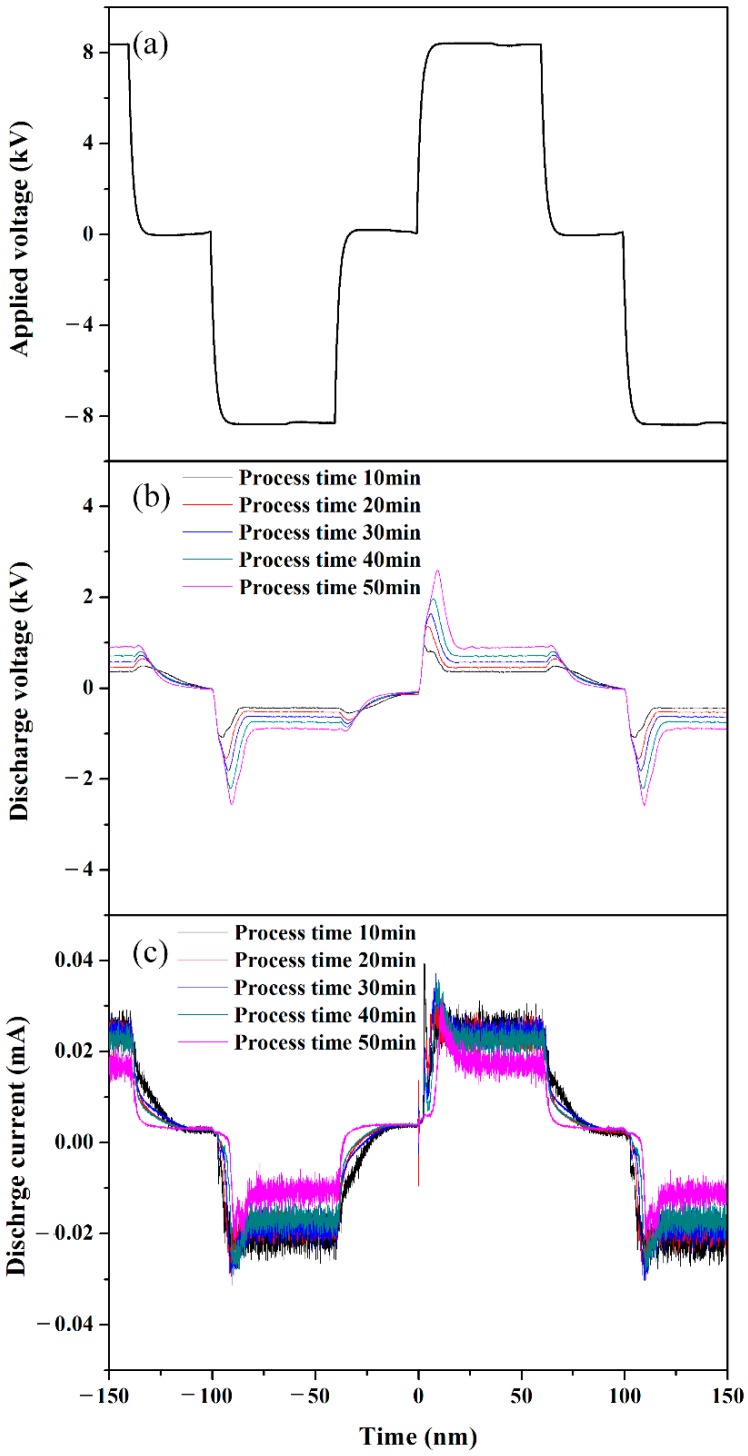
(**a**) Applied voltage, (**b**) discharge voltage, (**c**) discharge current measured during solution plasma process with gas bubble channel in liquid aniline monomer relative to process time.

**Figure 4 polymers-11-00105-f004:**
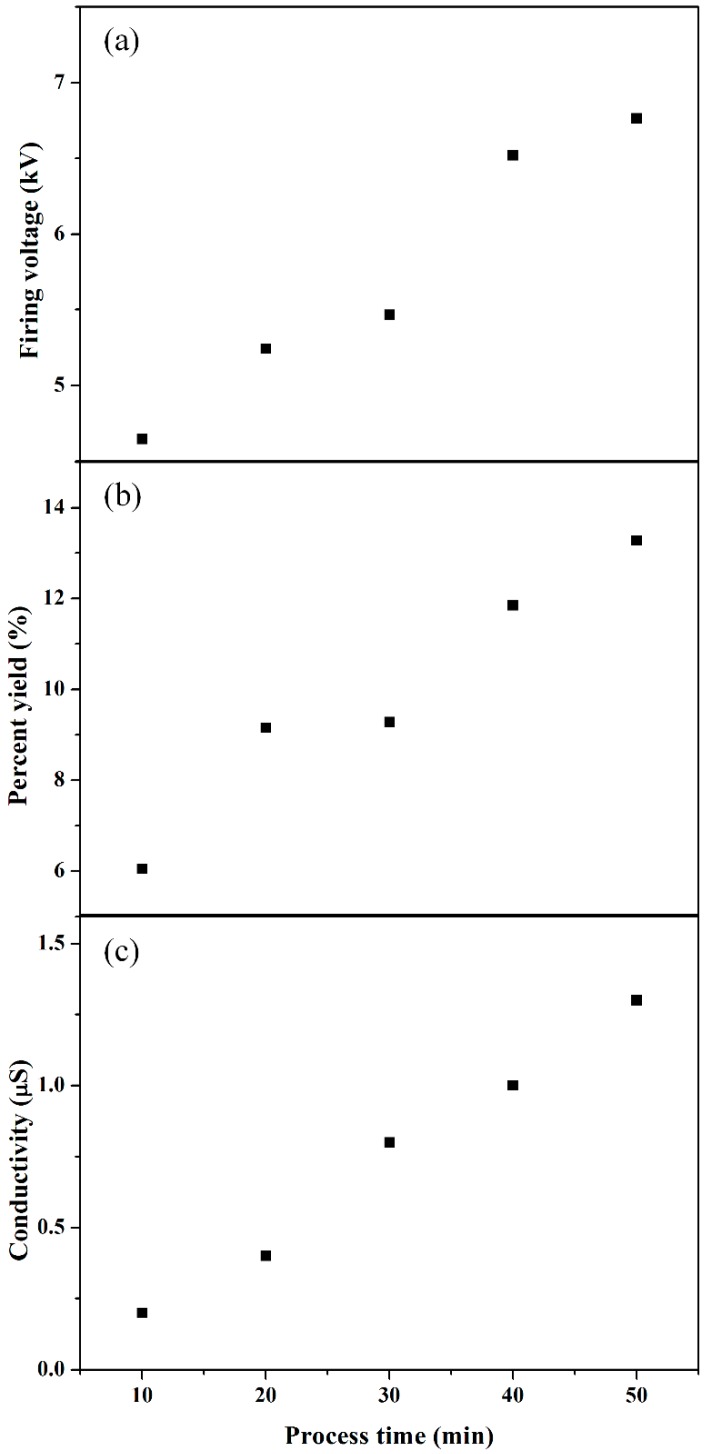
Variations of (**a**) firing voltage, (**b**) percent yield, and (**c**) conductivity of synthesized polyaniline (PANI) nanoparticle dispersed solution in liquid aniline monomer during process time up to 50 min.

**Figure 5 polymers-11-00105-f005:**
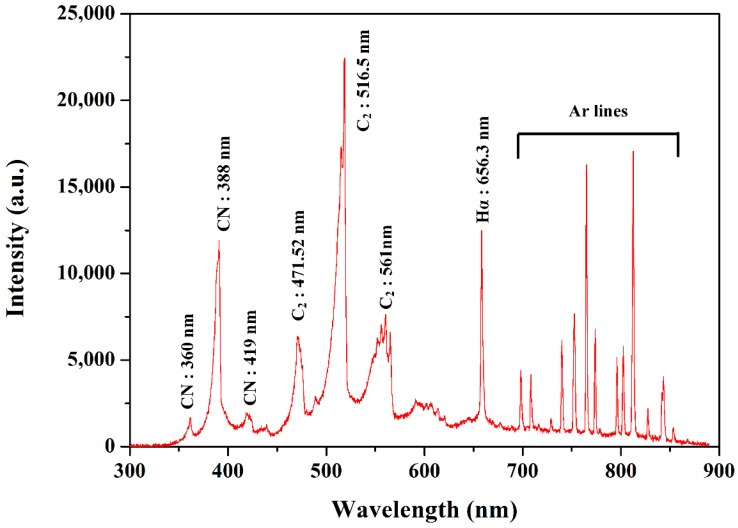
Optical emissions measured from plasma discharge in liquid aniline when PANI nanoparticles were synthesized by solution plasma process with gas bubble channel.

**Figure 6 polymers-11-00105-f006:**
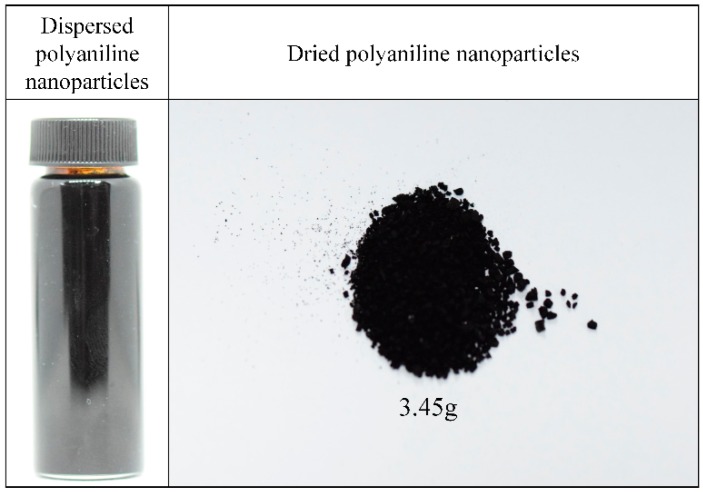
Dispersed PANI synthesized for 50 min and dried PANI nanoparticles.

**Figure 7 polymers-11-00105-f007:**
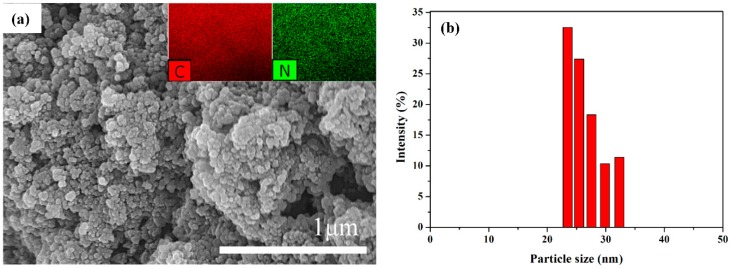
(**a**) Scanning electron microscope (SEM) image and energy dispersive X-ray spectroscopy (EDS), and (**b**) size distribution by dynamic light spectroscopy (DLS) of PANI nanoparticles synthesized by solution plasma with a gas bubble channel; insets of (**a**) indicate carbon and nitrogen measured by EDS.

**Figure 8 polymers-11-00105-f008:**
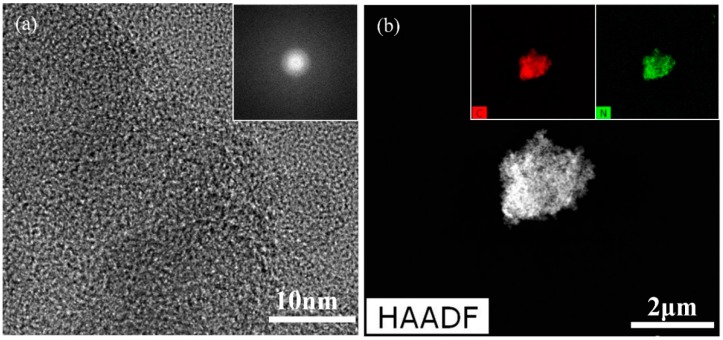
Transmission electron microscope (TEM) image and energy dispersive X-ray spectroscopy (EDS) and selected area electron diffraction (SAED) pattern of PANI synthesized by solution plasma with gas bubble channel. (**a**,**b**) TEM images with different magnifications; inset of (**a**) is SAED pattern, whereas insets of (**b**) are carbon and nitrogen elements using EDS.

**Figure 9 polymers-11-00105-f009:**
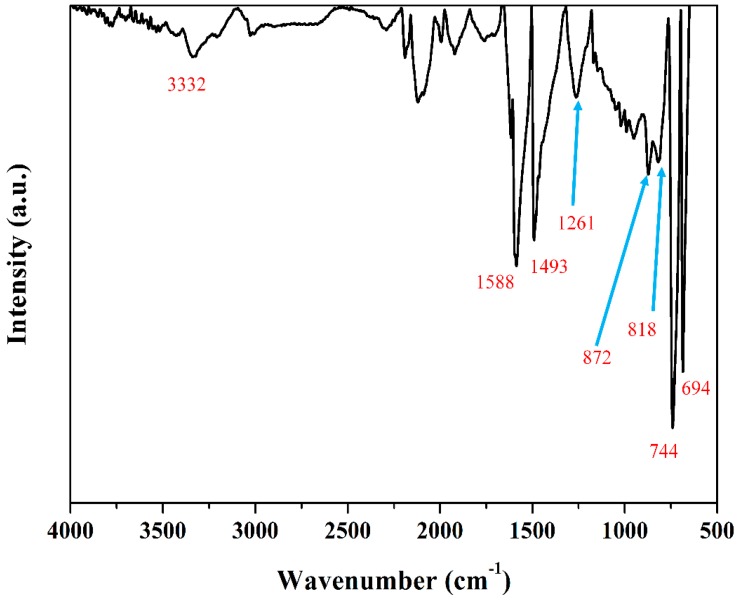
Fourier transformation infrared (FTIR) spectrum of synthesized PANI nanoparticles by solution plasma process with gas bubble channel.

**Figure 10 polymers-11-00105-f010:**
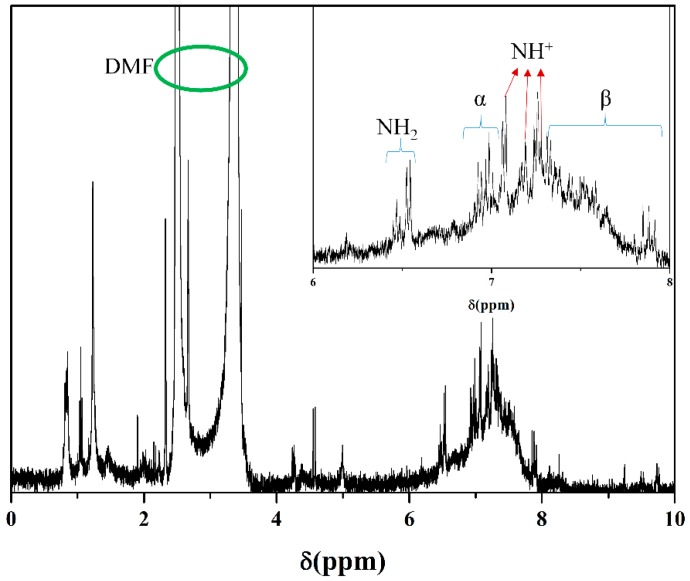
^1^H-nuclear magnetic resonance (NMR) spectrum of PANI nanoparticles synthesized by solution plasma with gas bubble channel, PANI nanoparticles were dissolved by deuterated dimethylformamide (DMF-d_7_) solvent.

**Figure 11 polymers-11-00105-f011:**
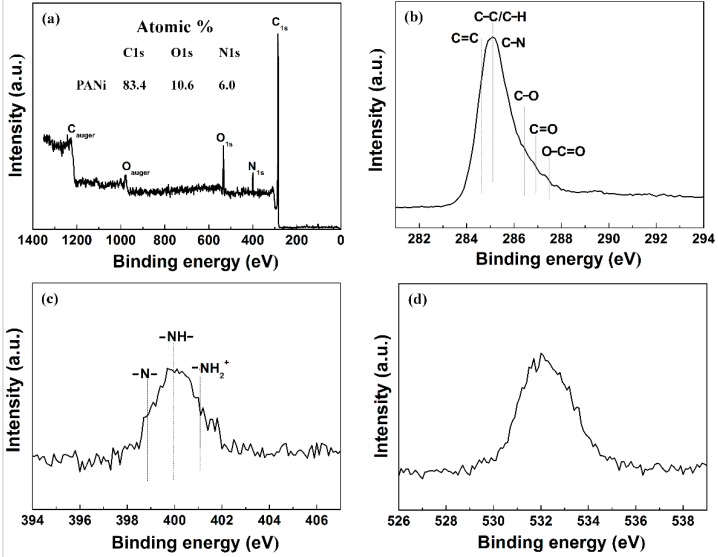
X-ray photoelectron spectra (XPS) of synthesized PANI nanoparticles by solution plasma with gas bubble channel (**a**) XPS wide scan spectra, (**b**) C 1s spectra, (**c**) N 1s spectra, and (**d**) O 1s spectra of synthesized PANI nanoparticles.

**Table 1 polymers-11-00105-t001:** Measured PANI nanoparticles by gel permeation chromatography.

Samples	Mw (kDa)	Mn (kDa)	Polydispersive Index (PDI)
Aniline monomer	0.093	-	-
PANI	9.7059	8.1299	1.19

**Table 2 polymers-11-00105-t002:** Peak assignment and envelop composition of various C 1s core level spectra of PANI observed in X-ray photoelectron spectra (XPS) in Figure 11b.

**Sample**	**C 1s Peaks Assignment (eV) and Envelopment Composition (%)**					
	284.6	285.1	285.7	286.4	286.9	287.5
	C=C	C–C/C–H	C–N	C–O	C=O	O–C=O
PANI	25.4	38.7	17.2	10.7	4.3	3.7

**Table 3 polymers-11-00105-t003:** Peak assignment and envelop composition of various N 1s core level spectra of PANI observed in X-ray photoelectron spectra (XPS) in Figure 11c.

**Sample**	**N 1s Peaks Assignment (eV) and Envelopment Composition (%)**		
	398.7	399.7	401.1
	–N–	–NH–	–NH_2_–
PANI	31.0	52.6	16.4

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
