# Peer review of "Synthesis of a Polyaniline Nanoparticle Using a Solution Plasma Process with an Ar Gas Bubble Channel"

_polymers, 2019, doi:10.3390/polym11010105_

Round 1

Reviewer 1 Report

Nicely written and well presented results. There are only two comments I had regarding the results which I feel must be addressed before publishing:

In the OES data the line at 390nm is attributed to CH but I think that this is likely the CN peak at 388nm. The side bands of the CN band are present and thus the central mode should be present as well. CH could be there as well but would need to be examined with higher resolution.

In the XPS data the peak positions are noted with two decimal places of precision. The precision of the instrument is not likely high enough to state positions to this level. One decimal place is what is usually used.

 Very few corrections to be made grammatically:

25 change the word "the" to "a"

195 change "strong" to "strongly"

253 the word nuclear is misspelled

Probably a few other small edits that I missed.

Author Response

Revised manuscript number MS #Polymers-414815 entitled “Synthesis of Polyaniline Nanoparticle using Solution Plasma Process with Ar Gas Bubble Channel” by J.-G. Shin et al.

First of all, the authors really appreciate the reviewers’ valuable comments for the paper. Based on the reviewers’ comments, the descriptions for the experimental results are clarified in the revised manuscript. In addition, the related explanations and discussions are compensated and intensified in the revised manuscript. As a result, the optical emission spectroscopy spectra in Figure 5 and the X-ray photon spectroscopy results in Fig. 11 with Tables 2 and 3 are modified, and the dynamic light scattering for investigating size distribution in the Figure 7b is newly provided to express the experimental data more clearly as per the reviewers’ recommendations. Total figures and tables changed are given as follows.

      Old Manuscript

Revised   Manuscript

Fig.   1

Fig.   1

Fig.   2

Fig.   2

Fig.   3

Fig.   3

Fig. 4

Fig.   4

Fig. 5

Fig. 5 [Modified]

Fig. 6

Fig.   6

Fig. 7

Fig.   7a

Fig. 7b [NEW]

Fig. 8

Fig. 8

Fig.   9

Fig.   9

Fig.   10

Fig.   10

Fig.   11a

Fig. 11a [Modified]

Table   1.

Table   1.

Table   2.

Table 2. [Modified]

Table   3.

Table 3. [Modified]

I. Upon the reviewer #1’ comments

Nicely   written and well presented results. There are only two comments I had   regarding the results which I feel must be addressed before publishing:

► We appreciate your kind and detailed assessment of the work presented in this paper.

1.  

In   the OES data the line at 390nm is attributed to CH but I think that this is   likely the CN peak at 388nm. The side bands of the CN band are present and   thus the central mode should be present as well. CH could be there as well   but would need to be examined with higher resolution.

► The authors appreciate the in-depth comments from the reviewer. According to the reviewer’ comment, the assignment of the line at 388 nm in Fig. 5 is changed from ‘CH’ to ‘CN’, and the related sentences are also modified in the revised manuscript in order to avoid potential confusion.

Fig. 5 [Modified]

At lines 222 and 223 on page 7 in Results and Discussions:

The CN peaks of 388 nm are intense peaks, which are related to benzene ring of liquid aniline monomer. The CN violet system has two smaller peaks of 360 and 419 nm, indicating to amine group attached to benzene ring.”

2.  

In   the XPS data the peak positions are noted with two decimal places of   precision. The precision of the instrument is not likely high enough to state   positions to this level. One decimal place is what is usually used.

► We appreciate your valuable comment. According to the reviewer’ comment, all two decimal places in the XPS data of Fig. 11a, Tables 2, and 3 are modified to one decimal place.

Figure 11a [Modified]

Table 2. Peak assignment and envelop composition of various C 1s core level spectra of PANI observed in X-ray photoelectron spectra (XPS) in Figure 11b.

sample

C1s peaks assignment   (eV) and envelopment composition (%)

284.6

285.1

285.7

286.4

286.9

287.5

C=C

C-C/C-H

C-N

C-O

C=O

O-C=O

PANI

25.4

38.7

17.2

10.7

4.3

3.7

Table 2. [Modified]

Table 3. Peak assignment and envelop composition of various N 1s core level spectra of PANI observed in X-ray photoelectron spectra (XPS) in Figure 11c.

sample

N1s peaks assignment   (eV) and envelopment composition (%)

398.7

399.7

401.1

-N-

-NH-

-NH2-

PANI

31.0

52.6

16.4

Table 3. [Modified]

At lines 312 and 313 on page 14 in Result and Discussions

Atomic percentage of PANI nanoparticle is 83.4% for C 1s, 6.0% for N 1s, and 10.6% for O 1s.

At lines 315 to 320 on page 14 in Result and Discussions

In Figure 11b, six carbon-containing component peaks of C 1s at 284.1, 284.6, 285.7, 286.4, 286.9, and 287.5 eV are confirmed to be corresponding to C-C/C-H, C=C, C-N, C-O, C=O and O=C-O, respectively. Figure 11c shows the N 1s spectra of the PANI nanoparticles where the three peaks centered at 398.7, 399.7, and 401.1 eV correspond to the quinoid imine (-N-), benzenoid amine-like (-NH-) structure, and positively charged nitrogen (NH2+) [45, 46], respectively.

3.  

Very   few corrections to be made grammatically:

► We appreciate your kind and detailed assessment of the work presented in this paper. According to the reviewer’ comment, some words, sentences, and phrases are modified to improve the English expression including grammar of our paper.

At lines 30-32 on page 1 in Introduction:

Many scientists or engineers have investigated many considerable advances and changes in the synthesis and characterization of conducing polyaniline (PANI) for the last sixty years [1-7].

At lines 156-158 on page 4 in Results and Discussions:

“The distance between two capillary glass tubes for injection of Ar gas is 3 mm. In the case that the distance between two capillary glass tubes is more than 3 mm, Ar gas bubble channel from two capillary glass tubes is difficult to form in discharge region.

At lines 158 and 159 on page 4 in Results and Discussions:

Applied peak voltage, bipolar rectangular pulse width, frequency are 16.4 kV, 60 μs, and 5 kHz, respectively.

At lines 201 and 202 on pages 5 in Results and Discussions:

This phenomenon confirms that the plasma discharge in liquid aniline monomer strongly depends on the formation of gas bubble channel between two tungsten electrodes.

At line 209 on page 7 in Results and Discussions:

Figure 5 shows the optical emissions measured from plasma discharge in liquid aniline

At lines 218-220 on page 7 in Results and Discussions:

Molecular emission peaks of CN, CH, and C2 are produced by the electron impact dissociation of liquid aniline monomer.

At lines 223 on page 7 and 224 on page 8 in Results and Discussions:

The CN violet system has two smaller peaks of 360 and 419 nm, indicating to amine group attached to benzene ring.

At lines 252 and 253 on page 10 in Results and Discussions:

The insets in Figure 7a are the results of EDS, which confirm that the PANI nanoparticles involve components of carbon and nitrogen.

At lines 268-271 on page 10 in Results and Discussions:

The EDS analysis in insets of Figure 8b shows that the PANI nanoparticles synthesized have carbon and nitrogen elements, as shown in insets of Figure 7, because PANI nanoparticles are composed of benzene ring, quinoid ring and amine group.

At line 261 on page 11 in Results and Discussions:

Figure 10. 1H-nuclear magnetic resonance (NMR) spectrum of PANI nanoparticles synthesized

Reviewer 2 Report

Comments:

The 1st sentence in the Introduction has some language issue. We usually say “Scientists” or “Engineers” have investigated …… for sixty years. Therefore, according to the authors, the subject “the last sixty years” represent a type of researchers? Please rewrite this sentence.

One possible factor may affect your experimental result, the distance between tungsten electrodes. It may cause the variation of nanoparticle quality and properties due to the repeatedly use of electrodes. Wear out or erosion of electrodes can happen. Therefore the inter-electrode distance will get longer and longer.

Nanoparticle size can be estimated from TEM measurement. Please provide the size distribution of synthesized PANI nanoparticles.

In the last sentence of the “Results and Discussion” section, the naming of organoamino functional groups is chaotic. What is “bamine”? NH2+ is not the symbol of atoms! Please follow the IUPAC system of nomenclature.

In tables 2 and 3, please add the unit used for XPS peaks.

Author Response

Revised manuscript number MS #Polymers-414815 entitled “Synthesis of Polyaniline Nanoparticle using Solution Plasma Process with Ar Gas Bubble Channel” by J.-G. Shin et al.

First of all, the authors really appreciate the reviewers’ valuable comments for the paper. Based on the reviewers’ comments, the descriptions for the experimental results are clarified in the revised manuscript. In addition, the related explanations and discussions are compensated and intensified in the revised manuscript. As a result, the optical emission spectroscopy spectra in Figure 5 and the X-ray photon spectroscopy results in Fig. 11 with Tables 2 and 3 are modified, and the dynamic light scattering for investigating size distribution in the Figure 7b is newly provided to express the experimental data more clearly as per the reviewers’ recommendations. Total figures and tables changed are given as follows.

      Old Manuscript

Revised   Manuscript

Fig.   1

Fig.   1

Fig.   2

Fig.   2

Fig.   3

Fig.   3

Fig. 4

Fig.   4

Fig. 5

Fig. 5 [Modified]

Fig. 6

Fig.   6

Fig. 7

Fig.   7a

Fig. 7b [NEW]

Fig. 8

Fig. 8

Fig.   9

Fig.   9

Fig.   10

Fig.   10

Fig.   11a

Fig. 11a [Modified]

Table   1.

Table   1.

Table   2.

Table 2. [Modified]

Table   3.

Table 3. [Modified]

II. Upon the reviewer #2’ comments

1.  

The   1st sentence in the Introduction has some language issue. We usually say   “Scientists” or “Engineers” have investigated …… for sixty years. Therefore,   according to the authors, the subject “the last sixty years” represent a type   of researchers? Please rewrite this sentence.

► We appreciate your kind and detailed assessment of the work presented in this paper. According to the reviewer’s comment, this sentence in Introduction is modified in the revised manuscript based on your comment.

At lines 30-32 on page 1 in Introduction:

Many scientists or engineers have investigated many considerable advances and changes in the synthesis and characterization of conducing polyaniline (PANI) for the last sixty years [1-7].

2.    

One   possible factor may affect your experimental result, the distance between   tungsten electrodes. It may cause the variation of nanoparticle quality and   properties due to the repeatedly use of electrodes. Wear out or erosion of   electrodes can happen. Therefore the inter-electrode distance will get longer   and longer.

► The authors appreciate the in-depth comments from the reviewer. In conventional method, discharge without gas bubble channel is very difficult to be produced in liquid aniline monomer. In addition, most particles and materials synthesized using streamer and unstable discharge without gas bubble channel tend to show poor nanoparticle quality and properties due to the production of wear out or erosion of electrodes. However, in this experimental, when adopting the gas bubble channel, the chemical element compositions (wt%) of tungsten on PANI nanoparticles (from EDS results) were remarkably reduced from 6.36% to 1.37% compared to no gas bubble channel. These results confirm that the solution plasma process with gas bubble channel can reduce erosion of the electrodes. In addition, in the case that the distance between two tungsten electrodes is more than 1 mm, initial discharge voltage is very high and discharge is difficult to form in discharge region. Therefore, according to the reviewer’ comment, the following sentences are newly provided in ‘Results and Discussions’ at lines 146-148 and 150-152 on page 4 and at line 254-258 on page 10 in the revised manuscript.

At lines 146-148 on page 4 in Results and Discussions:

In addition, most particles and materials synthesized using streamer and unstable discharge without gas bubble channel tend to show poor nanoparticle quality and properties due to the production of wear out or erosion of electrodes.

At lines 150-152 on page 4 in Results and Discussions:

“The distance between two tungsten electrodes is 1 mm. In the case that the distance between two tungsten electrodes is more than 1 mm, initial discharge voltage is very high and discharge is difficult to form in discharge region.”

At lines 254-258 on page 10 in Results and Discussions:

“From EDS results, when adopting the gas bubble channel, the chemical element compositions (wt%) of tungsten on PANI nanoparticles were remarkably reduced from 6.36% to 1.37% compared to no gas bubble channel. These results confirm that the solution plasma process with gas bubble channel can produce uniform PANI nanoparticles with very small size and can reduce erosion of the electrodes.”

3.  

Nanoparticle   size can be estimated from TEM measurement. Please provide the size   distribution of synthesized PANI nanoparticles.

► The authors appreciate the in-depth comments from the reviewer. According to the reviewer’ comment, the size distribution using dynamic light scattering (DLS) analysis in Fig. 7b is newly added in the revised manuscript, and also the following sentences are newly provided in the revised manuscript.

Figure 7b. [NEW]

At lines 19-23 on page 1 in Abstract:

To investigate the characteristics of synthesized PANI nanoparticles, field emission scanning electron microscopy, dynamic light scattering, transmission electron microscopy, selective area electron diffraction (SAED) pattern, Fourier transform infrared spectroscopy (FTIR), gel permeation chromatography, 1H-nuclear magnetic resonance (1H-NMR), and X-ray photo spectroscopy (XPS) are examined.”

At lines 25 and 26 on page 1 in Abstract:

The result indicates that PANI nanoparticles have a spherical shape with the size between 25-35 nm.

At lines 109-113 on page 3 in Materials and Methods:

2.7 Dynamic Light Scattering

The size distributions and average particle sizes of PANI nanoparticles ranging from 0.6 nm to 10 μm are determined by dynamic light scattering (DLS, Otsuka Electronics Co. Ltd., Osaka, Japan). Before the measurement of size distribution, the dispersions of PANI nanoparticles are prepared by 20 min sonication in ethanol.”

At lines 244-247 on page 9 in Results and Discussions:

Figure 7. (a) Scanning electron microscope (SEM) image and energy dispersive X-ray spectroscopy (EDS), and (b) size distribution by dynamic light spectroscopy (DLS) of PANI nanoparticles synthesized by solution plasma with gas bubble channel; insets of (a) indicate carbon and nitrogen measured by EDS.

At lines 250-252 on page 10 in Results and Discussions:

The size distribution results in Figure 7b show that the nano sizes of PANI nanoparticles are ranged from 25 to 35 nm with a narrow size distribution.

At lines 254-258 on page 10 in Results and Discussions:

From EDS results, when adopting the gas bubble channel, the chemical element compositions (wt%) of tungsten on PANI nanoparticles were remarkably reduced from 6.36 to 1.37% compared to no gas bubble channel. These results confirm that the solution plasma process with gas bubble channel can produce uniform PANI nanoparticles with very small size and can reduce erosion of the electrodes.

At line 328 on page 14 in Conclusions:

The SEM, DLS, and TEM confirms that size of PANI nanoparticle is estimated to be about tens of nanometer and has spherical shape.”

4.    

In   the last sentence of the “Results and Discussion” section, the naming of   organoamino functional groups is chaotic. What is “bamine”? NH2+ is not the   symbol of atoms! Please follow the IUPAC system of nomenclature.

► The authors appreciate the in-depth comments from the reviewer. According to the reviewer’ comment, the related sentence is modified to express the experimental data more clearly in the revised manuscript.

At line 319 on page 14 in Results and Discussions:

Figure 11c shows the N 1s spectra of the PANI nanoparticles where the three peaks centered at 398.7, 399.7, and 401.1 eV correspond to the quinoid imine (-N-), benzenoid amine-like (-NH-) structure, and positively charged nitrogen (NH2+) [45, 46], respectively.

5.

In tables 2 and 3, please   add the unit used for XPS peaks.

► The authors appreciate the in-depth comments from the reviewer. According to the reviewer’ comment, units in Tables 2 and 3 are newly provide in the revised manuscript.

Table 2. Peak assignment and envelop composition of various C 1s core level spectra of PANI observed in X-ray photoelectron spectra (XPS) in Figure 11b.

sample

C1s peaks assignment   (eV) and envelopment composition (%)

284.6

285.1

285.7

286.4

286.9

287.5

C=C

C-C/C-H

C-N

C-O

C=O

O-C=O

PANI

25.4

38.7

17.2

10.7

4.3

3.7

Table 2. [Modified]

Table 3. Peak assignment and envelop composition of various N 1s core level spectra of PANI observed in X-ray photoelectron spectra (XPS) in Figure 11c.

sample

N1s peaks assignment   (eV) and envelopment composition (%)

398.7

399.7

401.1

-N-

-NH-

-NH2-

PANI

31.0

52.6

16.4

Table 3. [Modified]
